# Motor Cortex Stimulation Reversed Hypernociception, Increased Serotonin in Raphe Neurons, and Caused Inhibition of Spinal Astrocytes in a Parkinson’s Disease Rat Model

**DOI:** 10.3390/cells10051158

**Published:** 2021-05-11

**Authors:** Ana Carolina P. Campos, Miriã B. Berzuíno, Gabriela R. Barbosa, Helena M. R. C. Freire, Patricia S. Lopes, Danielle V. Assis, Erich T. Fonoff, Rosana L. Pagano

**Affiliations:** 1Division of Neuroscience, Hospital Sírio-Libanês, São Paulo 01308-060, SP, Brazil; anacarol.pcampos@gmail.com (A.C.P.C.); miriabenatti@gmail.com (M.B.B.); garocha23@gmail.com (G.R.B.); helenafreire97@gmail.com (H.M.R.C.F.); patriciasanae@hotmail.com (P.S.L.); dani.varin@gmail.com (D.V.A.); 2Department of Neurology, Division of Functional Neurosurgery, University of Sao Paulo Medical School, Sao Paulo 012469-03, SP, Brazil; erich.fonoff@gmail.com

**Keywords:** motor cortex, epidural stimulation, pain control, Parkinson’s disease, serotonin, glial cells, spinal cord

## Abstract

Persistent pain is a prevalent symptom of Parkinson’s disease (PD), which is related to the loss of monoamines and neuroinflammation. Motor cortex stimulation (MCS) inhibits persistent pain by activating the descending analgesic pathways; however, its effectiveness in the control of PD-induced pain remains unclear. Here, we evaluated the analgesic efficacy of MCS together with serotonergic and spinal glial modulation in an experimental PD (ePD) rat model. Wistar rats with unilateral striatal 6-OHDA and MCS were assessed for behavioral immobility and nociceptive responses. The immunoreactivity of dopamine in the substantia nigra and serotonin in the nucleus raphe magnus (NRM) and the neuronal, astrocytic, and microglial activation in the dorsal horn of the spinal cord were evaluated. MCS, without interfering with dopamine loss, reversed ePD-induced immobility and hypernociception. This response was accompanied by an exacerbated increase in serotonin in the NRM and a decrease in neuronal and astrocytic hyperactivation in the spinal cord, without inhibiting ePD-induced microglial hypertrophy and hyperplasia. Taken together, MCS induces analgesia in the ePD model, while restores the descending serotonergic pathway with consequent inhibition of spinal neurons and astrocytes, showing the role of MCS in PD-induced pain control.

## 1. Introduction

Parkinson’s disease (PD) is a progressive and complex neurodegenerative disease that presents with bradykinesia, akinesia, and gait balance, mainly due to severe loss of dopaminergic neurons within the nigrostriatal pathway [1,2,3]. Nonmotor symptoms, including psychiatric disturbances (such as depression and anxiety), autonomic failures (such as constipation), sleep disorders, and sensory dysfunctions (such as olfactory deficit and pain), occur in PD patients before the clinical motor symptoms that characterize the disease [4,5]. Persistent pain is one of the major contributors to the deterioration of quality of life in PD patients [6,7,8,9,10]; this is because, among other reasons, the pain syndrome in PD is commonly underestimated, possibly due to the absence of specific tools for accurate and reliable diagnosis and classification and due to inadequate analgesic therapy [11]. Nonsteroidal anti-inflammatory drugs and opioids widely used in persistent pain treatment do not seem to have a significant effect on PD patients [12,13,14]. Dopamine replacement in PD patients and experimental models has shown positive results in decreasing pain sensitivity [15,16,17]. However, long-term dopaminergic therapy can lead to undesirable effects, such as motor fluctuations during off-periods of the drug, dyskinesia, and pain hypersensitivity, which are associated with disease progression and drug exposure; this, in turn, exponentially worsens quality of life [18,19,20]. Beside dopamine, serotonin deficit also plays a major role in PD pathophysiology and is related to nonmotor symptoms such as pain and depression [21,22,23,24,25]. In rodent PD model, a deficiency in the descending analgesic pathways results in opioidergic deficit, glial activation, and neuronal hyperexcitability in the dorsal horn of the spinal cord (DSHC), leading to increased central sensitization and consequent pain syndrome [17,25,26]. Hence, we hypothesized that therapeutic approaches based on the reinforcement of descending analgesic control may be more effective for pain management in PD patients.

Since treating pain in individuals with PD can be challenging, many efforts have been made to find an effective treatment for PD-induced pain syndrome. Deep brain stimulation (DBS) of the subthalamic nucleus or the internal globus pallidus is** an effective treatment option for advanced PD refractory to oral pharmacotherapy, with good response to motor and nonmotor symptoms, including pain syndrome [27]. However, the surgical procedure is complex, expensive, and offers risks to patients, such as perioperative, hardware-related complications and side effects [28,29,30]. There are also some contraindications for DBS, including levodopa-resistant motor symptoms, chronic immunosuppression, dementia, psychiatric disorders, and structural brain abnormalities [31,32]. Hence, less invasive neurostimulation procedures for patients unresponsive to or excluded from DBS have been proposed. Motor cortex stimulation (MCS) by transcranial magnetic stimulation (TMS), transcranial direct current stimulation (tDCS), and epidural motor cortex stimulation (eMCS) has been shown to improve pain hypersensitivity in neuropathic pain conditions [33,34,35,36,37]. eMCS-induced analgesia mediates the activation of the descending analgesic pathways, principally by modulating the serotonergic system [38,39,40,41,42]. The serotonergic neurons of the nucleus raphe magnus (NRM), localized in the rostral ventromedial medulla (RVM), are critical for descending analgesic control, and they are one of the main sources of serotonin (5-HT) to the DHSC [43,44]. In the DHSC, serotonin acting on 5HT_1A_ receptors inhibits nociceptive neurons and fibers by decreasing the release of excitatory neurotransmitters, such as glutamate and substance P [45,46]. Although several studies have suggested that MCS is effective in improving motor symptoms in PD patients [47,48,49,50,51,52] the efficiency of controlling PD-induced pain is still unclear [53] and the mechanism underlying its analgesic effect remains poorly understood. In this study, we aimed to understand whether eMCS improves the painful behavior in an experimental model of PD by modulating the descending serotonergic pathway and spinal cellular circuitry.

## 2. Methods

### 2.1. Experimental Design

Under stereotaxic conditions, rats were injected with striatal 6-OHDA (to induce the PD model) or saline (control). Nociceptive behavior was evaluated using the paw pressure test before surgery (basal measurement, BM). Seven days after the striatal injection, the animals were re-evaluated with the paw pressure test and assessed using the apomorphine-induced rotation test to confirm the dopaminergic deficit. Subsequently, transdural electrodes were implanted over the functional area of the left primary motor cortex associated with the hind limb. Fourteen days after the striatal injection, a group of animals was subjected to MCS for 15 min and, under stimulation, were evaluated in relation to behavioral immobility (with bar test) and nociceptive response (with paw pressure test). Animals not subjected to MCS were also evaluated. Four experimental groups were investigated: (1) animals with striatal saline + sham (only electrode implanted) (n = 7); (2) animals with striatal saline + MCS (stimulated) (n = 6); (3) animals with striatal 6-OHDA + sham (n = 6); (4) animals with striatal 6-OHDA + MCS (n = 8). One hour after the last behavioral test, the animals underwent transcardial perfusion, and their brains and spinal cords (L2–L5 segments) were collected for evaluation of immunoreactivity (IR) of tyrosine hydroxylase (TH) in the substantia nigra (SN) (to confirm the nigrostriatal lesion), 5-HT in the NRM, and c-Fos (neuronal activation marker), Iba-1 (microglial marker), and GFAP (astrocytic marker) in the DHSC (Figure 1A).

### 2.2. Animals

A total of 27 male Wistar rats (weighing 200–250 g) were used in this study. The rats were housed in acrylic boxes (three rats per box) for at least one week before the experimental procedures were initiated. The animals were maintained in appropriate rooms with controlled light/dark cycle (12/12 h) and temperature (22 ± 2 °C), with wood shavings and free access to water and rat chow pellets. All animal experiments were conducted and reported in accordance with the ARRIVE guidelines (http://www.nc3rs.org.uk/arrive-guidelines, accessed on 1 December 2020). The protocols used during the execution of this project were approved by the Ethics Committee on the Use of Animals at Hospital Sírio-Libanês (SP, BRA; protocol number CEUA 2009/06). Additionally, body weight of rats was analyzed weekly to confirm the wellbeing of the animals (Appendix A).

### 2.3. Surgical Procedure for PD Model Induction

The PD induction model was made as previously described [25,54]. Animals were anesthetized with isoflurane (4% induction, 2.5% maintenance in 100% oxygen) associated with local anesthesia (2% lidocaine, 100 μL/animal on the scalp). Under stereotaxic conditions, 12 µg of neurotoxin 6-hydroxydopamine (6-OHDA, Sigma-Aldrich, MO, USA) diluted in 2 µL of 0.9% saline with 0.2% ascorbic acid was injected at two different points into the left striatum (6 µg/µL of 6-OHDA at each point) [55]. The injection was performed using the Hamilton syringe at the following coordinates: +2.7 mm mediolateral and +4.5 mm dorsoventral (first point); +3.2 mm mediolateral, +0.5 mm anteroposterior, and +4.5 mm dorsoventral (second point), according to the rat brain atlas [56]. Animals injected with 1 µL of saline at two different points in the left striatum were used as controls. At the end of the injection, the needle was held in place for an additional 5 min to prevent backflow of the solution. After the striatal injection, animals were treated with non-steroidal anti-inflammatory drug (NSAID) (0.5 mg/kg, SQ, Meloxicam, Ourofino Pet, SP, BRA). The rats were returned to their home cages and monitored until complete recovery from anesthesia. The regular diet was supplemented with a dietary supplement (Ensure, Abbott, SP, BRA) once a day for three consecutive days to ensure full recovery of the animals after the nigrostriatal injury.

### 2.4. Evaluation of the Apomorphine-Induced Rotational Behavior

The apomorphine-induced rotation test was applied to validate the PD model, since the number of asymmetric rotations correlates with nigral degeneration, as previously demonstrated [17]. Rotational asymmetric behavior was evaluated using an automatic rotometer system (Rota-Count 8, Columbus Instruments, Columbus, OH, USA) 7 days after striatal saline or 6-OHDA injection. Animals were injected with the dopamine agonist apomorphine (1 mg/kg, s.c., Tocris Bioscience, BZ, Bristol, UK) dissolved in 0.9% saline and were evaluated over 30 min, as previously described [57]. The criterion for rotation was a 180° turn toward the side contralateral to the lesion. To reduce stress, the rats were habituated for 30 min one day before the rotational test. The saline rats were also evaluated.

### 2.5. Nociceptive Threshold Evaluation

The mechanical nociceptive threshold was evaluated using a pressure apparatus (EEF 440, Analgesimeter, Insight, SP, BRA) on the hind paws, as previously described [58]. Briefly, a force of increasing intensity (up to 16 g/s) was continuously applied to the back of the hind paws. The force (in grams) required to induce the withdrawal response of the pressed paw represented the nociceptive threshold. The nociceptive test was performed before the striatal injection of saline or 6-OHDA (basal measurement), 7 days after the striatal injection and 14 days after the striatal injection during the last minute of MCS. The results were analyzed by comparing basal and final measurements. The rats were habituated to the testing procedure one day before the experiment to minimize stress.

### 2.6. Surgical Procedure for Electrode Implant and MCS Protocol

After the rotational test, animals were submitted to the surgical procedures for electrode implantation in the primary motor cortex, as previously described [40,59]. Briefly, animals were deeply anesthetized with isoflurane (4% induction, 2.5% maintenance in 100% oxygen) associated with local scalp anesthesia (2% lidocaine, 100 μL/animal on the scalp). Under stereotaxic guidance, using a functional map developed by our group [60], a pair of transdural stainless steel electrodes (cylinders of 0.8 mm in diameter) was implanted over the left primary motor cortex in the corresponding area to the right hind paw. The coordinates considering bregma were +1.5 mm mediolateral and +1.0 mm anteroposterior dorsoventral according to the rat brain atlas [56]. The integrity of the dura mater was an absolute condition. Two fixation screws (implanted 4–6 mm away from the site of stimulation) and acrylic polymer were used to stabilize the implant and to ensure electrical isolation. A connector with the electrode pole was also fixed to the whole set. Electrical stimulation was applied to this set as previously described [61] in a single 15-min session (1.0 V, 60 Hz, and 210 μs; Medtronic electrical stimulator, Minneapolis, MN). The cathode was always chosen to be the posterior contact of the electrode because, according to the functional map, the site has a greater surface area corresponding to the hind limb [60]. The chosen MCS protocol provides better specificity for M1 stimulation in conscious nonanesthetized rats and is similar to clinically available. The sham group was subjected to the same conditions but did not receive stimulation. The rats were randomly divided into the sham and stimulated groups.

### 2.7. Evaluation of Behavioral Immobility

After 14 days of striatal injection, the animals during MCS were evaluated using the bar test to measure akinesia (typical catalepsy test). Sham animals treated with striatal saline or 6-OHDA were also evaluated. The immobility test involves placing an animal in an unusual posture and recording the time for the animal to correct this posture [62]. Behavioral immobility is characterized by muscle rigidity and failure to correct an imposed posture for a prolonged period. In this test, the animals were positioned with both forepaws on a 9-cm high horizontal bar (0.9 cm diameter). The time course during which the animal remained motionless in this imposed posture was considered the bar test elapsed time (with a cutoff of 120 s). The behavioral immobility endpoint was considered when both forepaws were removed from the bar or when the animal moved its head in an exploratory manner.

### 2.8. Immunohistochemistry

One hour after the last behavioral test (on the 14th day), the animals were anesthetized with ketamine/xylazine (0.5/2.3 mg/kg, i.p.) and then subjected to transcardial perfusion with 0.9% saline solution, followed by 4% paraformaldehyde (PFA) dissolved in 0.1 M phosphate buffer (PB, pH 7.4) for immunohistochemistry assay. The assay was performed as previously described [25,59]. Briefly, the brains and spinal cords were incubated with specific primary antibodies, mouse anti-TH (1:1000, MAB5280, Millipore, MA, USA), rabbit anti-5-HT (1:1000, NT-102, Protos Biotech, Vestland, Norway), mouse anti-GFAP (1:1000, G3893, Sigma-Aldrich, St. Louis, MO, USA), rabbit anti-Iba-1 (1:1000, 019-19741, Wako Chemicals, VA, USA), or rabbit anti-c-Fos (1:1000, ABE457, Millipore, MA, USA) and biotinylated secondary antibodies (1:200; Jackson ImmunoResearch, West Grove, PA, USA), and visualized with diaminobenzidine tetrahydrochloride (DAB, Sigma-Aldrich). Finally, images were captured using a light microscope (Eclipse E1000, Nikon, Melville, NY, USA). Quantification was performed using ImageJ software (National Institutes of Health, Bethesda, MD, USA; http://rsbweb.nih.gov/ij/; accessed on 1 December 2020). The positive cell count was realized for c-Fos and Iba-1, while the immunoreactivity (IR) was evaluated for TH, 5-HT and GFAP. The regions of interest, including the SN (from bregma −6.60 to −6.00), NRM (from bregma −10.56 to −11.88), and DHSC (L4–L6) were identified based on specific atlases [56].

To establish the positive cell count we manually counted the positive cells within DHSC from five different sections (30 um) per animal from four animals per group. For that, we used the “cell count” analyses in the ImageJ software (Appendix A). Then, the average of positive cell count from control animals (saline + sham) was normalized to 100% to compare the percentage of modulation in the other groups.

To establish the 5HT-IR in the NRM, and GFAP-IR in the DHSC we evaluated five different sections (30 um) per animal and from four animals per group. For that, we determined the threshold followed by the “analyze particles” plugin in the ImageJ software. We considered a median size of 2–3 mm^2^, and circularity of 0.25–1.0 (Appendix A). Then, the average of counts determined by the software of the control rats (saline + sham) was normalized to 100% to compare the percentage of modulation in the other groups.

The morphology of microglia was studied by determining the ramification index (RI), as described by Becker [63]. Briefly, the RI was quantified by dividing the area of a polygonal object that is defined by the cells’ most prominent projections by the cell area. Ramified, resting microglial cells have a large projection area and a relatively small cell area, while activated cells have both areas as nearly identical, with an RI of approximately 1.

### 2.9. Statistical Analysis

Sample size of the animals was established considering the paw pressure test as primary outcome [64], where the power (β) was considered as 80%. Results are expressed as mean ± standard error of mean (SEM). Data were analyzed using GraphPad Prism (San Diego, CA, USA), and statistical significance was assessed using one-way ANOVA (1-w-ANOVA), followed by Tukey’s multiple comparison post hoc tests, except for the apomorphine-induced rotation and paw pressure test, in which two-way ANOVA (2-w-ANOVA) was used, followed by Bonferroni’s multiple comparison post hoc tests. In all cases, statistical significance was set at *p* < 0.05.

## 3. Results

### 3.1. Validation of the Unilateral 6-OHDA-Induced PD Model

In an attempt to validate the PD model, apomorphine-induced rotational behavior was evaluated as the degree of nigral degeneration correlates with the number of asymmetric rotations, as previously shown [17]. Rats injected with unilateral striatal 6-OHDA presented asymmetric rotation to the contralateral side of the lesion (64.4 ± 2.3), when compared with control group of striatal saline-injected rats (27.6 ± 0.9) (2-w-ANOVA; F_(1,11)_ = 46.61, *p* < 0.0001, followed by Bonferroni’s *post-hoc* test; Figure 1B). All animals injected with striatal 6-OHDA showed an asymmetric rotational behavior. Seven animals were excluded from this study because they removed their implants before the final nociceptive tests; thus, the results presented correspond to five animals per group.

### 3.2. MCS Does Not Interfere with Dopamine Restoration in the SN but Improves PD Model-Induced Immobility and Hypernociception

Animals injected with striatal 6-OHDA showed a decrease in TH-IR in the SN *pars compacta* compared to the control animals, 14 days following the nigrostriatal lesion (Figure 1C,E). MCS did not change the TH-IR in either saline or 6-OHDA rats (Figure 1D,F). Rats subjected to the PD model demonstrated increased latency in the bar test (10.1 s ± 0.6) compared to the control group (3.3 s ± 0.5) (1-w-ANOVA; F_(3,11)_ = 10.49, *p* = 0.0038, followed by Tukey’s *post-hoc* test; Figure 2A), and MCS was able to reverse this phenomenon (2.5 s ± 0.3; Figure 2A). The unilateral striatal 6-OHDA decreased the nociceptive threshold of both hind paws (26.7 g ± 1.3), 7 and 14 days after the neurotoxin injection, compared to the control group (61.7 g ± 3.2) and the basal measurement (2-w-ANOVA; F_(3,24)_ = 114.4, *p* < 0.0001, followed by Bonferroni’s *post-hoc* test; Figure 2B,C). MCS increased the nociceptive threshold in both 6-OHDA (147 g ± 3.4) and saline animals (139.7 g ± 2.9) beyond the basal measurement (Figure 2B,C).

### 3.3. MCS Increases the Serotonin Production in the NRM in Lesioned Rats and Inhibits the PD Model Induced Spinal Hyperactivation

Animals injected with saline and subjected to MCS did not show any alteration of 5-HT-IR in the NRM compared to saline + sham rats. The 6-OHDA-lesioned rats presented a decrease in 5-HT-IR (57.3 ± 1.4) in the NRM compared to the control rats (100 ± 1.7), 14 days after lesion (1-w-ANOVA, F_(3,11)_ = 17.15, p = 0.0013, followed by Tukey’s *post-hoc* test; Figure 3A). Nevertheless, rats with striatal 6-OHDA subjected to MCS exhibited an exacerbated increase in serotonin immunoreactivity (149 ± 6.1) compared to other experimental groups (Figure 3A). In the DHSC, the MCS did not interfere with the neuronal activation pattern in the saline animals; however, 6-OHDA animals showed an intensified increase in c-Fos-positive cells (164.7 ± 5.5) (a neuronal hyperactivation) compared to the control animals (100 ± 0.9) (1-w-ANOVA, F_(7,23)_ = 8.814, *p* = 0.0002, followed by Tukey’s *post-hoc* test; Figure 3B and Appendix A). MCS in the 6-OHDA rats was able to normalize the spinal neuronal activation pattern (66.1 ± 1.4) compared to nonstimulated 6-OHDA rats (Figure 3B and Appendix A).

### 3.4. MCS Decreases the Spinal Astrocytic Hyperactivation without Interfering with Microglial Hypertrophy/Hyperplasia

Animals injected with striatal 6-OHDA showed an increase in GFAP-IR (214 ± 1.6) (1-w-ANOVA, F_(7,23)_ = 70.75, *p* < 0.0001, followed by Tukey’s *post-hoc* test; Figure 4A and Appendix A) and Iba-1 positive cells (hyperplasia phenomenon) (168.8 ± 7.3) (1-w-ANOVA, F_(7,23)_ = 11.98, *p* < 0.0001, followed by Tukey’s *post-hoc* test; Figure 4B and Appendix A) compared to the control animals (100 ± 2.2 for GFAP-IR and 100 ± 4.5 for Iba-1 positive cells), 14 days after the lesion. MCS normalized the GFAP-IR (91.3 ± 2) in 6-OHDA rats (Figure 4A and Appendix A) but did not alter the Iba-1 positive cell count (227.6 ± 9.6) when compared with the control group (Figure 4B and Appendix A). Considering that the MCS did not interfere with the hyperplasia phenomenon, we evaluated the hypertrophy phenomenon, which is another way to investigate the pattern of activated microglia [63]. The RI was determined to evaluate the hypertrophy of microglia cells. The RI of 6-OHDA animals, regardless of whether they were subjected to MCS, was approximately 1 (±0.3), while that of saline animals, regardless of whether they were stimulated, was approximately 2.5 (±1.7) (Figure 4C). These findings showed that animals subjected to the PD model presented hypertrophic microglia in the DHSC when compared with the control group and that MCS did not interfere with microglial activation in the spinal cord (Figure 4C).

## 4. Discussion

Persistent pain is one of the most common nonmotor symptoms of PD and contributes to its complexity [7,8,65]. The gold standard treatment for PD is dopamine replacement, which improves motor and nonmotor symptoms, including pain [15,17,19]. However, long-term dopamine therapy induces severe side effects that diminish the quality of life in individuals with PD [18,66]. Thus, nonpharmacological strategies, such as MCS, have been proposed to attenuate PD symptoms [67]. In this study, we showed that one session of eMCS improves mobility and induces analgesia in hemiparkinsonian rats while concomitantly increasing 5-HT-IR in the NRM and decreasing the activation pattern of neurons and astrocytes in the DHSC, without interfering with microglial number and morphology. In preclinical models, the use of noninvasive neurostimulation, such as tDCS and TMS, has limitations regarding current density, electrical field, and placement of the electrodes, which decreases the specificity of stimulating the M1 [68,69]. Hence, in an attempt to specifically stimulate M1, we placed the transdural electrodes in the M1 corresponding to the hind paw [60], where the nociceptive threshold was assessed. Implantation of the electrode is pivotal for the specific stimulation of M1 and for the reliability of our results, considering that the MCS was performed in conscious non-anesthetized rats and it is known that classic inhalational anesthesia interferes with nociceptive threshold [70]. Although this method is more invasive than tDCS and TMS, eMCS has a relatively low degree of invasiveness, and it guarantees the specificity of electrical stimulation in the M1 in rats, as previously demonstrated [60].

First, we validated our PD model demonstrating the rotational asymmetric behavior induced by apomorphine after striatal 6-OHDA, which indicates a disbalance between dopaminergic terminals in the striatum [71] and correlates with dopaminergic loss in the SN [17]. Concerning dopamine in the SN, we showed that eMCS did not interfere with the immunoreactivity of this monoamine in saline or experimental PD animals. Indeed, to date, no treatment has been able to stop progressive dopaminergic loss in PD [72]. The M1 firing pattern is highly influenced by midbrain dopamine [73,74]. Direct dopaminergic projections are diminished in subcortical motor structures and in the M1 of individuals with PD [75,76,77]. PD subjects demonstrate decreased cerebral flow in the motor cortex after movement initiation [78] probably due to the output connections from the basal ganglia upon activation of the direct and indirect motor pathways [79]. Taken together, these data suggest that M1 receives erroneous firing from the motor neurocircuitry, which is consequently transmitted to the spinal cord and peripheral nervous system, contributing to the emergence and maintenance of the motor and nonmotor symptoms of PD [80]. Here, rats subjected to the PD model showed motor deficits due to increased immobility in the bar, which was reversed by eMCS. Our data corroborate previous findings regarding M1 stimulation and improvement of motor symptoms in PD [52,67,81]. Interestingly, pyramidal neurons from the M1 project into the subthalamic nucleus (STN) and spinal cord [82] and are activated by STN stimulation [83]. The strategy of stimulating the specific STN from M1 can be beneficial both by reproducing the effectiveness of STN stimulation without the adverse effects of nonspecific activation of different STN circuits and by using a less invasive treatment intervention [67,84]. Cioni [53] suggested that MCS may be an alternative to DBS as it improves levodopa-induced dyskinesia, painful dystonia, and motor fluctuations after adjustment of the optimal stimulation parameters.

Regarding hypernociceptive behavior, control animals injected with striatal saline submitted to eMCS presented increased nociceptive threshold beyond basal measurement, suggesting that M1 stimulation induces the activation of analgesic pathways even without any commitment of nuclei involved in pain control, reinforcing previous studies with naive conscious rats [59,61]. As paw pressure test depends on a evoked motor response, it is important to highlight that eMCS used do not interfere with the general motor behavior of rats [61]. Furthermore, in the behavioral immobility stimulated saline rats spent the same time in the bar as nonstimulated saline rats, suggesting that eMCS does not induce motor impairment. Consistent with previous findings [17,25,85], the unilateral rat PD model induced bilateral hypernociception. We previously demonstrated that the bilateral hypernociception induced by unilateral striatal 6-OHDA model is accompanied by expressive bilateral loss of the descending analgesic pathway and spinal opioid system together with glial activation in the DHSC [25]. It is important to highlight that our results are based upon one single nociceptive test which measures the evoked motor response related to hypernociceptive behavior. However, we have also previously shown that the striatal 6-OHDA model is not able to produce motor deficit [17]. Thus, even though our paw pressure test is reliable, it should be noted the limitation of observing a single hypernociceptive phenomenon. Furthermore, here we showed that eMCS inhibits the bilateral hypernociception behavior induced by the PD model. Interestingly, eMCS also increased the nociceptive threshold beyond the basal measurement in both paws. We suggest that this increased nociceptive threshold is due to an acute activation of the descending analgesic pathway as discussed below. Recently, Suppa et al. [86] showed that motor evoked potential in the M1 in response to peripheral nociceptive stimulation was decreased in PD subjects compared with healthy subjects, suggesting that the activation impairment in the motor cortex of PD individuals may directly affect the proper functioning of the analgesic circuitry. In patients and animal models of chronic pain, eMCS induces analgesia due to the activation of the descending analgesic system [38,40,87], with critical role of the descending serotonergic pathway [59,88,89,90]. Considering we previously showed that the unilateral striatal 6-OHDA induced PD model decreases serotonin label in the NRM, and that the protection of this nucleus of the neurotoxin effect prevents the DP model-induced hypernociception [25], we investigated whether serotonin also plays a role in the eMCS-induced analgesic effect in hemiparkinsonian rats.

We confirmed in our PD model that nigrostriatal lesion induces a serotonergic deficit in the NRM, as previously shown in patients [22,91] and PD model [25]. These data reinforce that a deficit in the descending serotonergic pathway contributes to hypernociception in PD [25,92,93]. Considering that epidural M1 stimulation is able to activate the descending analgesic pathway in clinical trials for patients with persistent pain [38,39], we suggest that MCS may be effective for pain management in PD, although further clinical studies should be performed. Our group previously showed that eMCS increased 5-HT-IR in the NRM in healthy conscious animals [59]. In the present study, we demonstrated that eMCS is also able to increase the 5-HT-IR in the NRM of hemiparkinsonian rats above the basal level when compared with control animals, suggesting that the eMCS in the presence or absence of persistent pain facilitates the activation of descending serotonergic pathway. Considering the serotonergic deficit observed in lesioned rats, the eMCS induces an overcompensation mechanism in the NMR in the attempt to control the hypernociception induced by the 6-OHDA model.

The DHSC is a major structure of nociceptive control that receives monoaminergic projections from the descending analgesic pathway and modulates nociceptive input through inhibitory and excitatory mediators [94]. Here, we showed a spinal hyperactivation, observed by increase of c-Fos staining in the superficial laminae in the DHSC of hemiparkinsonian rats. We have previously shown that the PD model induces an increase in c-Fos-positive cells, accompanied by a decrease in opioidergic modulation [17]. Consistent with our findings, persistent pain in a rat model similar to the one presented in this study was associated with hyperexcitability of nociceptive neurons in the DHSC observed by electrophysiology [26]. Interestingly, eMCS was able to reverse the c-Fos-positive cells to control levels, suggesting an inhibition in nociceptive neuron activation, as previously shown in naive rats [59]. Nevertheless, further methodology strategies need to be addressed to confirm the inhibition of nociceptive output after eMCS in PD model. Although serotonin-induced analgesia has been widely attributed to the inhibition of nociceptive neurons within the DHSC [45,95], its role in the modulation of spinal astrocytes and microglia has been discussed [96,97,98,99]. PD model-induced hypernociception has been associated with an increase in classical activated astrocytes and microglia in the spinal cord [17,25]. In this study, we observed that hemiparkinsonian animals showed an increase in the GFAP-IR, both increased number of cells and increased cell volume, consistent with activated astrocyte [100]. Notably, the eMCS decreased the GFAP-IR, which is related to astrocyte reactivity, both number and morphology. However, by immunohistochemistry images, we can observe a clear distinction in relation to morphology of astrocytes, while the number of positive cells was less affected in stimulated 6-OHDA rats when comparing with nonstimulated 6-OHDA rats. Taken together these data suggest that a single session of eMCS is able to inhibit the hypertrophy without interference with hyperplasia of reactive astrocytes in the DHSC. Interestingly, serotonin improves calcium signaling in astrocytes and secretion of antioxidant molecules [101,102]. In this sense, the increased production of serotonin in the NRM of lesioned rats submitted to MCS may consequently have increased the release of these monoamines in the DHSC, thereby inhibiting the classical proinflammatory activation of astrocytes. It is important to highlight that the control of hyperactivation of astrocytes in the DHSC is causally related to the inhibition of persistent hypernociception in rodent models [103]. Corroborating with the importance of spinal inflammation to increased level of nociception in the PD model, we observed the phenomena of hyperplasia and hypertrophy of Iba-1-positive cells in the DHSC in nonstimulated lesioned rats. Curiously, eMCS was unable to inhibit microglial hyperactivation in the DHSC, neither the cell number nor shape of these cells. It was shown in vitro that serotonin enhances microglial injury-induced motility by activating the serotonin receptor, where phagocytic activity was decreased without changing the ramification index [104]. Thus, we can suggest that eMCS by stimulating the serotonergic system could interfere with microglial activation; however, the state of classical activation or alternative cannot be observed by investigating the ramification index. Of note, monoaminergic projections into the DHSC, such as dopaminergic and noradrenergic, are involved in the inhibition of nociceptive neurons as well as in the induction of anti-inflammatory mechanisms regarding the inhibition of glial cells [105,106,107,108,109,110]. Considering that serotonin, dopamine, and noradrenalin also mediate MCS-induced analgesia [41,42,88], it is important to highlight that these monoamines may act jointly with serotonin in the control of glial and neuronal activation in the spinal cord, in response to cortical stimulation in the PD model.

Here, we showed that a single session of eMCS activates the descending serotonergic system, which probably leads to increased serotonin release in the DHSC with consequent inhibition of spinal nociceptive neurons and astrocytes, without interfering with microglial hyperactivation (Figure 5). To provide prolonged and persistent analgesia in PD, MCS must be applied continuously, as normally occurs in the clinical practice, to establish a more expressive glial inhibition in different areas involved with the nociceptive system. Nevertheless, beyond the scope of this preclinical study, effort must be made to execute double-blind placebo-controlled randomized clinical trials in an attempt to standardize stimulation protocols to investigate the therapeutic benefits of cortical stimulation in PD. Optimal behavioral, neuroimaging, and biomolecular assays should be conducted to further comprehend the underlying effectiveness of MCS in controlling the motor and nonmotor symptoms of PD.

## 5. Limitations of Study

This work sheds light into the concept that MCS would be able to aid individuals with PD suffering from persistent pain by activating the descending analgesic pathway and improving the spinal nociceptive modulation. However, this is a preclinical study evaluating hyperalgesia nociceptive behavior in rats submitted to a chemical PD model. Hence, further studies should be addressed in PD individuals to validate the efficacy of MCS in PD pain. Moreover, specific brain targets in a noninvasive setting are challenging in rodents and therefore, to ensure the M1 specificity, here we used epidural MCS. Less invasive techniques, such as tDCS and TMS, should also be investigated in PD individuals with persistent pain. We strongly believe that, considering our findings and the prior literature, double-blind randomized trials should be performed to verify MCS efficacy and improve the quality of life of these individuals.

## 6. Conclusions

Our results suggest that MCS reverses the bilateral hypernociception of rats subjected to the unilateral PD model while restores the striatal neurotoxin-induced serotonergic deficit in the NRM. The increase in serotonin output, with consequent release in the spinal cord, may be related to a decrease in the activation pattern of astrocytes and neurons, but not microglia, in the DHSC. Considering the beneficial effects of the eMCS on persistent pain, this stimulation protocol could be considered an optional tool for improving the painful behavior in PD patients after careful investigation in clinical settings.

## Figures and Tables

**Figure 1 cells-10-01158-f001:**
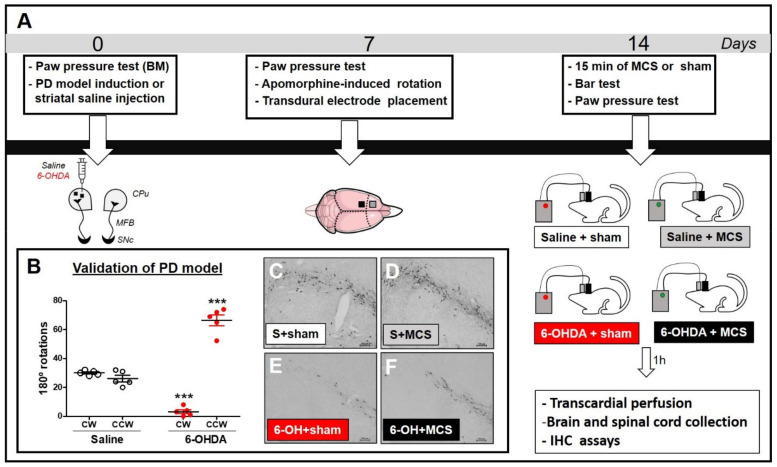
Experimental design (**A**). Animals were evaluated in the paw pressure test, anesthetized, and injected with 6-OHDA (to induce the experimental PD model, ePD) or saline (control group) in the left striatum. Animals were re-evaluated in the paw pressure test and apomorphine-induced rotation test 7 days after striatal injections to validate the ePD model. Next, animals underwent implantation of the transdural electrode in the motor cortex. Fourteen days after the PD model induction, animals were submitted to motor cortex stimulation (MCS) or sham for 15 min and re-evaluated in the paw pressure test. One hour after the last nociceptive evaluation, the brains and spinal cords were collected for immunohistochemistry (IHC) assays. Control animals, whether submitted to MCS or not, were also evaluated. The immunoreactivity of tyrosine hydroxylase (a marker of dopaminergic neurons) in the substantia nigra (SN), serotonin in the nucleus raphe magnus (NRM), and c-Fos, GFAP, and Iba-1 in the dorsal horn of the spinal cord (DHSC) were evaluated by IHC. Evaluation of apomorphine-induced rotation (**B**) to confirm dopaminergic deficit. Values represent the mean + SEM. *** *p* < 0.001 when compared to the control group (saline). Photomicrographs of immunoreactivity for TH (**C**–**F**) of rat saline + sham (S + sham, (**C**)), rat saline + MCS (S + MCS, (**D**)), rat 6-OHDA + sham (6-OH + sham, (**E**)), and rat 6-OHDA + MCS (6-OH + MCS, (**F**)). 6-OHDA: 6-hydroxydopamine; CW: clockwise; CCW: contra-clockwise.

**Figure 2 cells-10-01158-f002:**
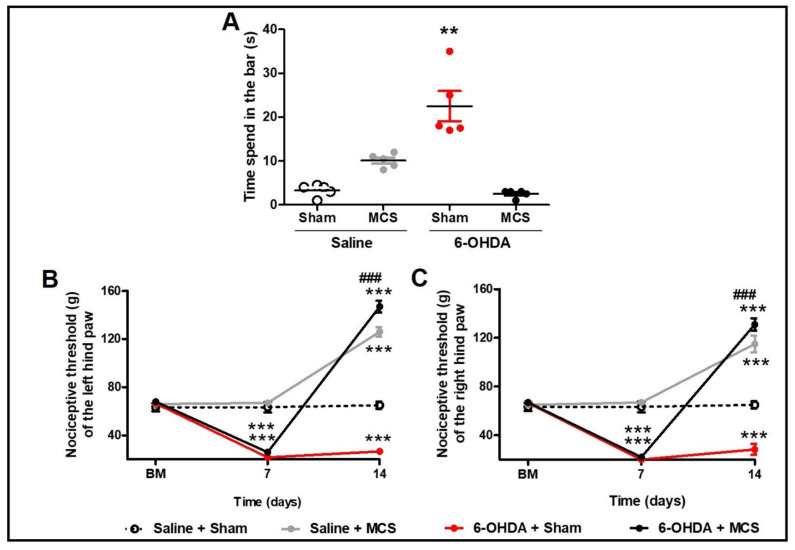
Motor and nonmotor symptoms. Rats were evaluated in relation to behavioral immobility in the bar test (**A**) 14 days after the striatal injection. The values represent the mean ± SEM (n = 5 per group). ** *p* < 0.01 when compared to saline + sham group (1-w-ANOVA followed by Tukey post-hoc test). Rats were evaluated in relation to nociceptive threshold in the paw pressure test in the left (**B**) and right (**C**) hind paws. Nociceptive behavior was evaluated before (basal measurement, BM), and 7 and 14 days after striatal injection. Fourteen days after the striatal injection, the motor cortex stimulation (MCS) was applied in a single 15-min session (1.0 V, 60 Hz, and 210 μs). The values represent the mean ± SEM (n = 5 per group). ** *p* < 0.01; *** *p* < 0.001 when compared to saline + sham group; ^###^
*p* < 0.001 when compared to 6-OHDA + sham group (2-w-ANOVA followed by Bonferroni’s post-hoc test).

**Figure 3 cells-10-01158-f003:**
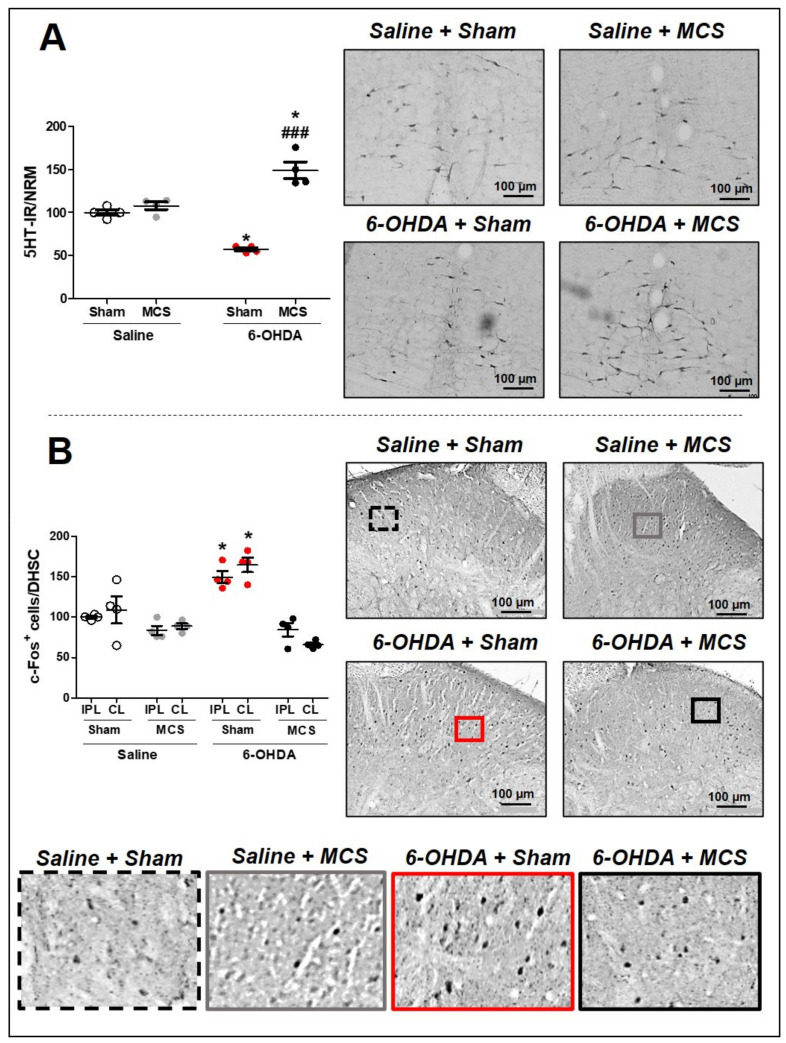
Serotonin immunoreactivity in the nucleus raphe magnus (NRM). Rats were evaluated in relation to immunoreactivity (IR) for serotonin (5-HT) in the NRM (**A**). Values represent the mean + SEM normalizing saline + sham as 100%. * *p* < 0.05 when compared to the control group (saline + sham); ^###^
*p* < 0.001 when compared to the 6-OHDA + sham group (1-w-ANOVA followed by Tukey post-hoc test). Photomicrographs of 5-HT-IR of rat saline + sham, rat saline + MCS, rat 6-OHDA + sham, and rat 6-OHDA + MCS. Pattern of activation in the dorsal horn of the spinal cord (DHSC). Rats were evaluated in relation of c-Fos-positive cells in the DHSC (**B**). Values represent the mean + SEM normalizing saline + sham as 100% (n = 4 per group). * *p* < 0.05 when compared to the control group (saline + sham) (1-w-ANOVA followed by Tukey post-hoc test). Photomicrographs of c-Fos-positive cells of rat saline + sham, rat saline + MCS, rat 6-OHDA + sham, and rat 6-OHDA + MCS. Enhanced photomicrographs for better labeling visualization. 6-OHDA, 6-hydroxydopamine; MCS, motor cortex stimulation; IPL, ipsilateral to the lesion; CL, contralateral to the lesion.

**Figure 4 cells-10-01158-f004:**
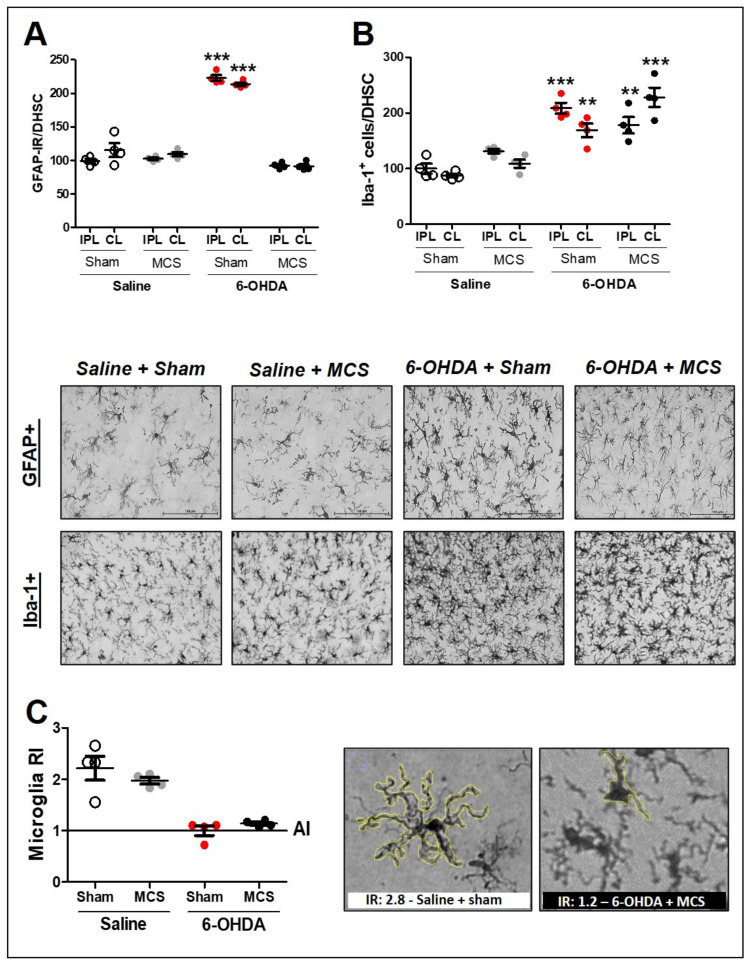
Glial cells in the spinal cord. Astrocytes in the dorsal horn of the spinal cord (DHSC) (**A**). Rats were evaluated in relation to immunoreactivity (IR) for GFAP in the DHSC (**A**). Values represent the mean + SEM normalizing saline + sham as 100% (n = 4 per group). *** *p* < 0.001 when compared to the control group (saline + sham) (1-w-ANOVA followed by Tukey post-hoc test). Photomicrographs of GFAP-IR of rat saline + sham, rat saline + MCS, rat 6-OHDA + sham, and rat 6-OHDA + MCS. Microglia in the DHSC (**B**,**C**). Rats were evaluated in relation to number of Iba-1 positive cells (hyperplasia—(**B**)). Values represent the mean + SEM normalizing saline + sham as 100%. ** *p* < 0.01; *** *p* < 0.001 when compared to the control group (saline + sham). Ramification index (RI) of microglial cells (**C**). Values represent the mean + SEM where values similar to 1 indicate an amoeboid morphology. Photomicrographs demonstrating the morphology difference between saline + sham and 6-OHDA + MCS. 6-OHDA, 6-hydroxydopamine; AI, activation index; MCS, motor cortex stimulation; IPL, ipsilateral to the lesion; CL: contralateral to the lesion.

**Figure 5 cells-10-01158-f005:**
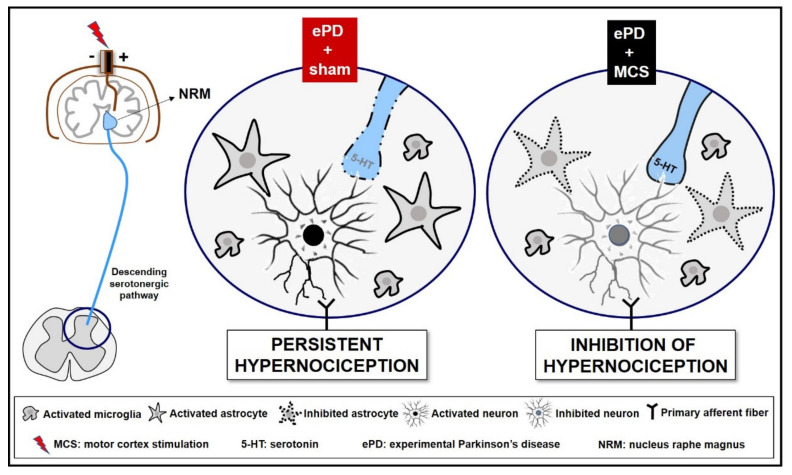
Representative scheme descending analgesic control in an experimental Parkinson’s disease (ePD) model comparing animals treated with motor cortex stimulation (MCS) or sham. The striatal 6-OHDA, beyond the progressive dopaminergic loss in the nigrostriatal pathway, decreased the serotonin immunoreactivity in the nucleus raphe magnus (NRM). This serotonergic nucleus projects to the dorsal horn of the spinal cord (DHSC) to control the ascendant nociceptive inputs. However, probably due to loss of serotonergic efferences in the spinal cord, the ePD induced an increase in neuronal activation, and increased immunoreactivity of astrocytes and microglial hyperplasia and hypertrophy in the DHSC, a phenomenon responsible for the maintenance of hypernociception pain. The MCS was able, not only to restore, but also increase the serotonin immunoreactivity in the NRM, without change in dopamine loss induced by ePD. In addition, the MCS decreased the neuronal activation pattern and the immunoreactivity of astrocytes in the DHSC, without interfering with the ePD-induced microglial hyperplasia and hypertrophy. Taken together, we suggest that MCS-induced analgesia is partially due to restoration of the descending serotonergic pathway resulting in the increase in spinal serotonin release, with consequent inhibition of the hyperactivation of spinal neurons and astrocytes in hemiparkinsonian rats. Different monoamines, such as dopamine and noradrenaline, play an important role in pain control and may be also related to the MCS-induced analgesia in ePD rats.

## Data Availability

The data presented in this study are available on request from the corresponding author.

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
