# Peer review of "Motor Cortex Stimulation Reversed Hypernociception, Increased Serotonin in Raphe Neurons, and Caused Inhibition of Spinal Astrocytes in a Parkinson’s Disease Rat Model"

_cells, 2021, doi:10.3390/cells10051158_

Round 1

Reviewer 1 Report

The Authors addressed in general the points asked in the review

Author Response

To Reviewer #1 of Cells,

Reviewer #1: The Authors addressed in general the points asked in the review.

Response: We thank the reviewer for all the suggestions raised which have considerably improved our manuscript. Follow the certificate of English editing at the end of the file attached.

Reviewer 2 Report

Campos and colleagues investigated the effects of MCS on hyprnociception in PD rat model. The manuscript is well written and provides some novel insights in this field. I have some comments to help improve the quality of the manuscript.

  1. Page 5 the first paragraph, could the authors provide the rationale of choosing this particular type of MCS protocol and is this protocol similar to or identical to any clinically available eMCS?
  2. Please provide sample size justification and power. This study had a small sample size (n=5 in each group) with multiple comparisons and was likely underpowered.
  3. Please provide effect sizes of each outcome measure, especially the comparisons between active and sham eMCS in PD groups in the results section.
  4. Page 10 line 340: while there were effects on mobility and hypernociception, and changes in serotonin, activation pattern in neurons and astrocytes in active eMCS group, causal relationship cannot be inferred here. As the authors did not conduct causal mediation analysis and the study was likely underpowered. Please amend this sentence to avoid misleading the readers. Similarly, in the conclusion and the abstract.
  5. Page 11, line 416: while these data suggest targeting M1 with eMCS had some positive effects in PD rat model, and thus a possible avenue for treating persistent pain in PD, they do not infer this is effective for pain management for PD. Please amend this sentence.
  6. The limitations of this study should be acknowledged and discussed in the Discussion section. Could authors comment on the clinical implications of the findings? For example, what type of clinical MCS could be beneficial and investigated in the clinical PD populations?
  7. Page 13, line 505: the MCS used in this study was not “non-invasive”. Further, the data of this study do not support the claim that this treatment should be considered a capable tool for improving painful behaviour in patients with PD as this is a preclinical study with small sample size. Please amend this statement.
  8. Abbreviation SN should be spelled fully in the first appearance in the text. There is an error in Fig 1A: DP model should be PD.

Author Response

To Reviewer #2 of Cells,

We would like to thank the reviewer for the critical review of our manuscript and for the helpful suggestions provided. Below we detailed how we responded to each concern. In this new version of the revised manuscript, the changes are highlighted in yellow. We also inserted the certificate of English editing at the end of this file.

Campos and colleagues investigated the effects of MCS on hypernociception in PD rat model. The manuscript is well written and provides some novel insights in this field. I have some comments to help improve the quality of the manuscript.

- Page 5 the first paragraph, could the authors provide the rationale of choosing this particular type of MCS protocol and is this protocol similar to or identical to any clinically available eMCS?

Response: We discussed the rationale of this particular type of MCS in the first paragraph of the Discussion section: “In pre-clinical models, the use of non-invasive neurostimulation, such as tDCS and TMS, has limitations regarding current density, electrical field, and placement of the electrodes, which decreases the specificity of stimulating the M1. Hence, in an attempt to specifically stimulate M1, we placed the transdural electrodes in the M1 corresponding to the hind paw, where the nociceptive threshold was assessed. Implantation of the electrode is pivotal for the specific stimulation of M1 and for the reliability of our results, considering that the MCS was performed in conscious non-anesthetized rats and it is known that classic inhalational anesthesia interferes with nociceptive threshold. Although this method is more invasive than tDCS and TMS, eMCS has a relatively low degree of invasiveness, and it guarantees the specificity of electrical stimulation in the M1 in rats, as previously demonstrated.” Moreover, the eMCS protocol here used is similar to clinically available and it is widely used for neuropathic pain with good outcomes (Rasche et al., Pain; 121: 43-52; doi: doi: 10.1016/j.pain.2005.12.006.). We added a sentence in the Methodology section: The chosen MCS protocol provides better specificity for M1 stimulation in conscious non-anesthetized rats and is similar to clinically available, as suggested by the reviewer. We prefer to keep the full explanation of the rationale of choosing the eMCS protocol in the Discussion section to make the Methodology section more fluid.

- Please provide sample size justification and power. This study had a small sample size (n=5 in each group) with multiple comparisons and was likely underpowered.

Response: At our Institution the Animal Ethics Committee requires a sample size analyzes to determine the sample size considering the importance of the 3 Rs: Replacement, Reduction and Refinement. Therefore, we used a sample size calculation standardize for animals use (especially rodents) previously described (Charan et al. Pharmacol Pharmacother 2013; (4) 303–306. doi:10.4103/0976-500X.119726.)

Where: 2 SD2(1.96 + 0.842)2/d2

SD: standard deviation – 15.0 (from pilot studies from our group)

d = effect difference = difference between mean values – 40.0

With this sample size calculation, we achieve a number of 4.2 animals per group considering α = 5% and β = 80%.

Here, we used the paw pressure test analyzes as primary outcome. Hence, we added the power information in the Methodology section (Statistical analysis) in the revised manuscript: “Sample size of the animals was established considering the paw pressure test as primary outcome [64], where the power (β) was consider as 80%.”  

- Please provide effect sizes of each outcome measure, especially the comparisons between active and sham eMCS in PD groups in the results section.

Response: We added the values of mean ± standard error of mean (SEM) of each outcome measure in the Result section in the revised manuscript.

- Page 10 line 340: while there were effects on mobility and hypernociception, and changes in serotonin, activation pattern in neurons and astrocytes in active eMCS group, causal relationship cannot be inferred here. As the authors did not conduct causal mediation analysis and the study was likely underpowered. Please amend this sentence to avoid misleading the readers. Similarly, in the conclusion and the abstract.

Response: We agree with the reviewer and changed the sentences in the Abstract, Discussion and Conclusion sections in the revised manuscript to avoid the misleading information of causal relationship between the data available.

- Page 11, line 416: while these data suggest targeting M1 with eMCS had some positive effects in PD rat model, and thus a possible avenue for treating persistent pain in PD, they do not infer this is effective for pain management for PD. Please amend this sentence.

Response: We agree with the reviewer that we cannot infer the eMCS effectiveness for pain management for PD patients. However, eMCS has been shown to be effective in neuropathic patients due to improve the descending analgesic pathway in clinical trials. Therefore, we changed the sentence to make it clearer in the fourth paragraph of the Discussion section: Considering that epidural M1 stimulation is able to activate the descending analgesic pathway in clinical trials for patients with persistent pain [38,39], we suggest that MCS may be effective for pain management in PD, although further clinical studies should be performed.”

- The limitations of this study should be acknowledged and discussed in the Discussion section. Could authors comment on the clinical implications of the findings? For example, what type of clinical MCS could be beneficial and investigated in the clinical PD populations?

Response: We added a Limitation of study section in the revised manuscript where we stated that: This work sheds light into the concept that MCS would be able to aid individuals with PD suffering from persistent pain by activating the descending analgesic pathway and improving the spinal nociceptive modulation. However, this is a pre-clinical study evaluating hyperalgesia nociceptive behavior in rats submitted to a chemical PD model. Hence, further studies should be address in PD individuals to validate the efficacy of MCS in PD pain. Moreover, specific brain targets in a non-invasive setting are challenging in rodents and therefore, to ensure the M1 specificity, here we used transdural MCS. Less invasive techniques, such as tDCS and TMS, should also be investigated in PD individuals with persistent pain. We strongly believe that, considering our findings and the prior literature, double-blind randomized trials should be performed to verify MCS efficacy and improve the quality of life of these individuals.”

- Page 13, line 505: the MCS used in this study was not “non-invasive”. Further, the data of this study do not support the claim that this treatment should be considered a capable tool for improving painful behaviour in patients with PD as this is a preclinical study with small sample size. Please amend this statement.

Response: We appreciate the reviewer suggestion, and amended the last sentence of the Conclusion section bringing to the reader’s attention the importance to investigate the efficacy of MCS in clinical settings. Considering the beneficial effects of the eMCS in persistent pain, this stimulation protocol could be considered an optional tool for improving the painful behavior in PD patients after careful investigation in clinical settings”.

- Abbreviation SN should be spelled fully in the first appearance in the text. There is an error in Fig 1A: DP model should be PD.

Response: We would like to thank the reviewer for bring these errors to our attention. Substantia nigra was fully spelled in Experimental design, when it first appears; and the figure 1A was fixed.

Reviewer 3 Report

In this study, the authors investigated the analgesic effect of motor cortex stimulation (MCS) on 6-OHDA-induced Parkinson disease in Wistar rats by detecting the immunoreactivity of serotonin in NRM and neuronal c-Fos expression as well as activation of astrocyte and microglia in the spinal dorsal horn. This study is interesting. However, the authors need to address the following questions before being ready for publication.

Major

  1. Figure 2B shows that even in the saline group, MCS significantly increased the paw mechanical threshold, which was much higher than the baseline. Please explain.
  2. By measuring the paw mechanical threshold, the authors found that MCS reduced the pain of Parkinson's disease caused by 6-OHDA, and the mechanism was due to the increase of serotonin in the NRM and the decrease of neuronal activity marker, c-Fos expression. However, in the saline group, MCS also changed the pain behavioral threshold, but had no effect on the release of serotonin and the expression of c-Fos. Please explain.
  3. In addition, Figures 3 and 4 provide the statistical results of bilateral c-Fos, GFAP and Iba-1staining, but only show images of one side of the spinal dorsal horn. In addition, the authors may provide GFAP and Iba-1 stained images of the entire dorsal horn instead of partial spinal slices.
  4. The authors mentioned in the discussion "the eMCS decreased the GFAP-IR, which was clearly observed in relation to morphology of astrocytes and not in relation to number of positive cells", however, the effect of MCS on the changes of number and morphology of astrocyte cells was not clearly distinguished in the result description.
  5. Fos protein is usually expressed early after drug injection. However, the results showed that a large amount of c-Fos expression was still observed on the 14th day after saline injection. Please explain.

Minor

There are some typos in the manuscript, such as “DP model” in Figure 1.

Author Response

To Reviewer #3 of Cells,

We would like to thank the reviewer for the critical review of our manuscript and for the helpful suggestions provided. Below we detailed how we responded to each concern. In this new version of the revised manuscript, the changes are highlighted in yellow. We also inserted the certificate of English editing at the end of this file.

In this study, the authors investigated the analgesic effect of motor cortex stimulation (MCS) on 6-OHDA-induced Parkinson disease in Wistar rats by detecting the immunoreactivity of serotonin in NRM and neuronal c-Fos expression as well as activation of astrocyte and microglia in the spinal dorsal horn. This study is interesting. However, the authors need to address the following questions before being ready for publication.

- Figure 2B shows that even in the saline group, MCS significantly increased the paw mechanical threshold, which was much higher than the baseline. Please explain.

Response: We agree with the reviewer that this is an interesting but surprising result. However, these findings were previous shown by other investigators. We explained the possible reasons for this behavior in the third paragraph of the Discussion section: “Regarding hypernociceptive behavior, control animals injected with striatal saline submitted to eMCS presented increased nociceptive threshold beyond basal measurement, suggesting that M1 stimulation induces the activation of analgesic pathways even without any commitment of nuclei involved in pain control, reinforcing previous studies with naive conscious rats [59,61].” These results suggest that eMCS possible activated the descending analgesic pathway in an acute manner, which explains the increased nociceptive threshold in our saline rats, similar to that observed in naive rats from the references provided.

- By measuring the paw mechanical threshold, the authors found that MCS reduced the pain of Parkinson's disease caused by 6-OHDA, and the mechanism was due to the increase of serotonin in the NRM and the decrease of neuronal activity marker, c-Fos expression. However, in the saline group, MCS also changed the pain behavioral threshold, but had no effect on the release of serotonin and the expression of c-Fos. Please explain.

Response: This is a very good question. We believe that under healthy conditions the eMCS has a very discreet action on the neuronal activation, seen by proto-oncogene marking, and on intracellular neurotransmitter labeling, which cannot be observed by our experimental assays. However, in the face of a sustained pain and pathophysiological states, as the PD model, these changes are quite evident and could be detected by our analysis. Of this manner, we believe that the cortical stimulation normalize or reduce the peripheral and central sensitization present in pathophysiological conditions; and that in healthy environment occurs a fine and delicate control of nociceptive system.

- In addition, Figures 3 and 4 provide the statistical results of bilateral c-Fos, GFAP and Iba-1staining, but only show images of one side of the spinal dorsal horn. In addition, the authors may provide GFAP and Iba-1 stained images of the entire dorsal horn instead of partial spinal slices.

Response: We agree with the reviewer the importance of demonstrating the entire dorsal horn and both sides of the immunostaining. Hence, we added a supplemental figure 2 with these images.

- The authors mentioned in the discussion "the eMCS decreased the GFAP-IR, which was clearly observed in relation to morphology of astrocytes and not in relation to number of positive cells", however, the effect of MCS on the changes of number and morphology of astrocyte cells was not clearly distinguished in the result description.

Response: We appreciate the reviewer from bring to our attention the lack of information in the results. However, we did not perform the separate quantification of number and morphology of GFAP positive cells. When measuring immunoreactivity, we are investigating astrocyte reactivity, which considers number and morphology together. Hence, we changed the sentence in the discussion session to highlight the limitations of investigating immunoreactivity in astrocytes. “the eMCS decreased the GFAP-IR, which is related to astrocyte reactivity, both number and morphology. However, by immunohistochemistry images, we can observe a clear distinction in relation to morphology of astrocytes, while the number of positive cells was less affected in stimulated 6-OHDA rats when comparing with non-stimulated 6-OHDA rats”.

- Fos protein is usually expressed early after drug injection. However, the results showed that a large amount of c-Fos expression was still observed on the 14th day after saline injection. Please explain.

Response: In this case, c-Fos staining in the DHSC is highly related to activation of nociceptive neurons. Hence, the trigger for the activation of these neurons was the paw pressure test rather than the drug injection.

-  There are some typos in the manuscript, such as “DP model” in Figure 1.

Response: We would like to thank the reviewer for bring this typo to our attention. Figure 1 was fixed.

Round 2

Reviewer 3 Report

The authors have addressed my concerns. I have no further questions. 

This manuscript is a resubmission of an earlier submission. The following is a list of the peer review reports and author responses from that submission.

Round 1

Reviewer 1 Report

Campos and collegues are presented and interesting work about the role of motor cortex stimulation (MCS) in Parkinson disease induced pain control. The authors showed that MCS reversed hypernociception in the Parkinson´s disease model of 6-OHDA in rats reestablishing serotonergic system and reducing neuronal and astroglial activation. The study is very simple, but it is clear, well designed and written. Results are well presented and figures are explanatory and illustrative. However, several issues should be addressed:

  • Why the authors have chosen the Parkinson disease model injecting the 6-OHDA in the striatum instead in the forebrain medial bundle?
  • Some issues should be added in methods section to clarify the way in which the experiments have been performed:
  • 5. Nociceptive threshold evaluation should be explained in more detail, specially the way in which the data have been collected and analyzed
  • Quantification of the immunohistochemistry sections should be better explained, mentioning the level of the sections chosen and the corresponding regions analyzed. Moreover, the use of a stereological counting method would be more suitable and accurate.  
  • In figure 2, the red and black symbols are both defined as 6-OHDA+sham creating confusion. Furthermore, the results obtained with the right and left hind paw in the nociceptive threshold (B and C) are surprisingly the same, in spite of the lesion was produced just in the left hemisphere. How the authors could explain that fact?
  • In figure 3, it is surprising that 5-HT levels in the raphe nucleus in the 6-OHDA+MCS group reach higher levels with respect the physiological controls treated with saline, please explain that result in the discussion section. In Figure 3 legend, line 4: please, correct “6-OHDO” for 6-OHDA.
  • In figure 4, GFAP-IR in the 6-OHDA+MCS group showed values similar that obtained in the controls treated with saline (A), however in the photomicrographs that group (E) showed higher number of GFAP cells than in the controls (B, C). Please, explain this contradiction.

Reviewer 2 Report

The authors assessed effects of motor cortex stimulation in a rat model of Parkinson's Disease that relies on the local toxicity of striatal injection of 6-OHDA toxicity. The model does not address that PD is a systemic disease with strong involvement of peripheral sensory neurons. Therefore, results with this toxicity model have limited predictability/relevance for human PD associated pain and the authors must be must more cautious with the term "pain" and the overall conclusion. The limitations of the model in terms of PD-pain need to be discussed.

Pain is a human feeling, the term should not be used if nociception or nociceptive hypersensitivity is meant or measured in rodents. Please replace throughout. Also holds true for terms like "analgesic". In rodents it is antinociceptive

Nociception was assessed with a Randall Selitto pressure test which requires restraining/fixing of the rats and the test rather measures freeing attempts and movements but does not reliably measure mechanonociception. The test has substantial bias by withdrawal movements and it is no longer used in nociceptive research. It is outdated. The results are not reliable as a single nociceptive test. A minimum would be additional thermal nociceptive tests.

In addition, nociceptive results were obtained with only five rats per group. At least twice as many must be used for reliable data.

Apparently, motor cortex stimulation also increased withdrawal in the pressure test in control mice (saline and MCS) strongly suggesting that the stimulation reduced movements (escape attempts). But this does not tell about nociception. Randall Selitto does not tell about nociception for rodents with alterations of motor functions.

The behavioral data are not convincing

The quantitative data are all shown as bar charts with sem. This way gives the least possible information and hides biological samples sizes and variability. It is strongly suggested to replace bar charts with scatter plots (or box/scatter or bar/scatter) with mean and SD. Samples sizes must also be revealed in figure legends telling the number of mice per group and of sections per mouse.

The procedures of histology/immunofluorescence quantification need to be detailed in the methods sections (e.g. background subtraction, threshold setting, size inclusions/exclusions. Exemplary counting masks could be shown as suppl. Figures.

The abbreviations in the abstract should be reduced. It is difficult to read.

Figures: Please label the figures with the groups so that one has not to search the legend. Legends have no sample sizes and there is no info about statistics

I cannot see a difference in Figure 3D and E which would explain the huge quantitative differences shown in the bar chart in A. Also B and E look similar. Figure G to H do not reveal the cFOS differences shown in F. The images look all similar.

Higher levels of cFOS in the DH could also be interpreted as stronger nociceptive input. Without further stainings, one cannot interpret the result.

Overviews of microglia are missing.

The drawing in Figure 5 goes far beyond the presented data.

The Discussion is overly long and loses focus on what was shown in the present study.

The conclusions go far beyond the presented data.

The procedures represent substantial experimental stress to the animals with 2 surgeries and apomorphin test. Data showing the general well being during experiments such as daily body weights should be shown in a suppl. figure